# Euphemism Identification via Feature Fusion and Individualization

Submission Id: xxx*

## ABSTRACT

Euphemisms are indirect words to convey sensitive concepts. For instance, "ice" serves as a euphemism for the target keyword "methamphetamine" in illicit transactions. Euphemisms are widely used on social media and darknet marketplaces to evade moderation and supervision. Thus, euphemism identification which aims to map the euphemism to its secret meaning (target keyword) is a crucial task in ensuring social network security. However, this task poses significant challenges, including resource limitations due to the unavailable of annotated datasets and linguistic challenges arising from subtle differences in meaning between target keywords. Existing methods have employed self-supervised schemes to automatically construct labeled training data, addressing the resource limitations. Yet, these methods rely on static embedding methods that fail to distinguish between literal and euphemistic senses, leading to confusion between target keywords with similar meanings. In addition, we observe that different euphemisms in similar contexts confuse the identification results. To overcome these obstacles, we propose a feature fusion and individualization (FFI) method for euphemism identification. First, we reformulate the task as a cloze task, making it more feasible. Next, we develop a feature fusion module to capture both dynamic global and static local features, enhancing discrimination between different euphemisms in similar contexts. Additionally, we employ a feature individualization module to ensure each target keyword has a unique feature representation by projecting features into their orthogonal space. As a result, FFI can effectively identify subtle semantic differences between similar euphemisms that refer to target keywords with similar meanings. Experimental results demonstrate that our method outperforms state-of-the-art methods and large language models (GPT3.5, Llama2, mPLUG-Owl, etc.), providing robust support for its effectiveness.

## KEYWORDS

Euphemisms, Euphemism Identification, Social Network Security, Feature Fusion, Feature Individualization

## 1 INTRODUCTION

Euphemisms are indirect, mild words or expressions that have long been used in human communication to conceal secret information [2]. For example, they can be used to express politeness and avoid embarrassment when discussing taboo topics [5]. However, in social media or darknet marketplaces, where cybercrimes such as drug trafficking [11, 17] and arms trading [9] occur, euphemisms are often used to cover up illegal transactions and evade supervision. For instance, the euphemisms "coke" and "ice" in Table 1 were used as substitutes for target keywords "cocaine" and "methamphetamine" in the drug category of Table 2. These euphemisms can seem innocent and vague, making it challenging to trace illicit

**Table 1: Examples of sentences containing euphemisms.**

| **Example sentences** (euphemisms are in bold) |
| --- |
| 1. We had already paid $70 for some shitty **weed** from a taxi driver but we were interested in some **coke** and the cubans. |
| 2. For all vendors of **ice**, it seems pretty obvious that it is not as pure as they market it. |
| 3. Back up before I pull my **nine** on you. |

**Table 2: Part of the target words.**

| Category | Target keywords |
| --- | --- |
| **Drug** | Amphetamine; Cocaine; Ecstasy; Heroin; Marijuana; Methamphetamine; opium |
| **Weapon** | Gun; Carbine; Gatling; Rifle; Pistol |
| **Sexuality** | Breast; Sex; Nipple; Condom; Pornography |

transactions. Therefore, identifying the target keyword of a given euphemism, known as euphemism identification, is essential for improving content moderation and combating underground markets for social network security. However, euphemisms are continually evolving, making it difficult to maintain an up-to-date corpus for training the euphemism identification task. Furthermore, the subtle distinction in meaning between the target keywords (e.g., cocaine and marijuana) adds to the complexity of the task[33].

Existing methods have primarily focused on detecting whether words are used in a euphemistic sense, with techniques evolving from conventional natural language processing [15, 19, 33] to deep learning pre-training models [24, 35, 36]. However, these methods rely on static word embeddings or sentiment analysis, which cannot distinguish meanings with similar semantics. Moreover, these methods can only detect euphemisms but fail to identify them to the corresponding target keywords. Despite this, current few studies on euphemism identification only focus on obtaining context information of the euphemisms to identify them to the corresponding target keywords, disregarding subtle semantic distinctions among the target keywords and failing to address the linguistic challenge posed by target keywords with similar semantics.

Based on the above challenges and our findings, we explore two kinds of subtle semantic differences in euphemism identification, which can provide additional dimensions to efficiently recognizing euphemisms. 1) **Subtle semantic differences among euphemisms**: euphemisms in the same category typically manifest in comparable contexts. For instance, as shown in Table 1, Sentence 1 contains both the euphemisms "weed" and "coke", resulting in nearly identical contexts for the two euphemisms. Euphemism "weed" in Sentence 1 and euphemism "ice" in Sentence 3 exhibit similar semantic contexts concerning drug trafficking. These similar contexts of distinct euphemisms may mislead a model into

learning confusing information for misidentifying the euphemisms. 2) **Subtle semantic differences among target keywords**: the semantic differences of some target keywords in the same category are subtle. As shown in Table 2, the meanings of the target keywords are similar, such as "marijuana", "cocaine" and "heroin" in the drug category, "gun", "carbine" and "rifle" in the weapon category, and "sex" and "pornography" in the sexuality category. The subtle semantic differences among the target keywords may lead the optimization of model parameters to fluctuate in different directions, thus resulting in degrading the model's performance.

To address these issues, we propose a feature fusion and individualization method (denoted as "FFI"), which regards the euphemism identification task as a cloze task, simplifying the euphemism identification task and making it easier to realize and evaluate in terms of the task form. The FFI model employs a feature fusion module to mitigate the problem of subtle semantic differences among euphemisms with similar contexts, by extracting discriminative features with rich semantics, using both dynamic global context and static local information. Considering the influence of subtle semantic distinction among target keywords, FFI uses a feature individualization module to project features into the orthogonal space, extracting distinct individual features of each target keyword and eliminating common features. In this way, FFI can accurately differentiate euphemisms in similar contexts referring to target keywords with similar semantics. Experimental evaluation on benchmark Drug, Weapon, and Sexuality datasets shows that our method yields top-k identification accuracies that are 45-55% higher than the state-of-the-art baseline methods. Additionally, as the number of training samples increases, FFI boosts performance 83-250% higher than the baseline methods do. Compared with the recognition accuracy of the large language model GPT3.5, we are 25 percentage points and 4 percentage points higher on the Weapon and Sexuality datasets respectively, and 2 percentage points lower on the Drug dataset. Notably, the time consumption and cost of FFI are one-quarter and one-tenth of GPT3.5 respectively. The main contributions are as follows:

- To our knowledge, we are the first to convert euphemism identification into a cloze task and propose a novel framework to recognize euphemisms. The cloze task formally makes euphemism identification more feasible, requiring the model to learn context to select the correct option, which helps the model better understand the meaning of euphemisms.
- We utilized a feature fusion module to fuse dynamic global and static local information of euphemisms to obtain fusion features that are discriminative with rich semantics. Furthermore, we used a feature individualization module to project target keywords into their respective difference space, making it easier to distinguish target keywords with similar semantics.
- Experiments demonstrate the effectiveness of our model, which significantly outperformed the SOTA models and is comparable to large language models. As the size of the training data increased, our model boosted performance much higher than the current best model did, resulting in good generalization ability.

## 2 RELATED WORK

Although the study on euphemisms in linguistics dates back many years [3, 8, 27], computational research on euphemisms is a relatively recent area of investigation, primarily focused on euphemism detection and identification. Euphemism detection refers to detecting words in a euphemistic manner, while euphemism identification refers to identifying the secret meaning of each euphemism.

### 2.1 Euphemism Detection

Euphemism detection task has been studied under supervised, semi-supervised, and unsupervised learning frameworks, using conventional natural language processing techniques, deep learning, pre-trained models, and other algorithms. Yuan et al. [33] expanded Word2vec [20, 21] and analyzed word semantic differences in cross-corpus to detect dark jargons. Magu and Luo [19] obtained candidate euphemisms by using word embedding cosine distances combined with network analysis. Both works rely on static word embeddings, which do not distinguish different meanings of the same word. Felt and Riloff [7] used lexical and context sentiment analysis to classify phrases as euphemistic, dysphemistic, or neutral based on the work of Thelen and Riloff [29]. Gavidia et al. [10] used roBERTa[1] to detect euphemisms based on the changes of sentiment scores.

Compared with conventional natural language processing methods, pre-trained models based on deep learning have made breakthroughs in several tasks. Ke et al. [14] constructed a word-based pre-training model DC-BERT to detect Chinese jargons via cross-corpus similarity analysis. Zhu et al. [36] used a masked language model (MLM) of BERT [6] as a filter to find euphemism candidates. Zhu and Bhat [35] used SpanBERT[2] to rank euphemisms to detect euphemisms on the basis of Zhu et al. [36].

### 2.2 Euphemism Identification

To our knowledge, there are few existing works on euphemism identification. The most relevant work was reported by Zhu et al. [36], who first proposed euphemism identification task. They developed an unsupervised algorithm that uses the bag-of-words model to analyze euphemisms in their sentence-level context, identifying each euphemism to the corresponding target keyword. The work by Yuan et al. [33] focused on identifying the hypernyms of euphemisms rather than directly identifying the specific meanings of them. They identify "horse" as an illicit drug rather than heroin (drug is the hypernym of heroin).

In terms of the social use of euphemisms, morphs are a kind of euphemism. The euphemism identification task is related to morph resolution. Sha et al. [25] ranked the target candidates based on character-word and radical-character-word embeddings. You et al. [32] ranked target entities based on cross-document corpus similarity metrics. These approaches focused on Chinese morph resolution, which are mostly event-based and temporal. Generally, the task of euphemism identification is also related to the lexical meaning discovery of unknown words. Ishiwatari et al. [12] extracted both the local and global features of unknown words and obtained the description of them. Yi et al. [31] incorporated features such as

---

[1]https://huggingface.co/roberta-base
[2]https://huggingface.co/SpanBERT/spanbert-large-cased

**Masked Sentences**:  As far as we know he was still smoking [MASK] but that was it.

**Candidates**:          1. Heroin   2. Ecstasy   3. Marijuana   4. Cocaine    …   n. Opium

**Training / Validation**: Mask the target keyword          **Testing**:  Mask the Euphemism

**Figure 1: Description of euphemism identification. During training: the input is a sentence with the target keyword masked out and the output is the target keyword. During testing: the input is a sentence with the euphemism masked out and the output is the target keyword corresponding to the euphemism.**

pinyin and radicals to extract semantic information to generate slang paraphrases. Word sense discovery aims to understand the meaning of unknown words by generating defined sentences, however, these approaches do not capture subtle differences between a set of target keywords with similar semantics.

In summary, the existing research has the following deficiencies. (1) Related work such as morph resolution [25, 32] and Chinese slang interpretation [31], proposed joint semantic feature extraction methods for attributes such as pinyin and radicals, which are unique attributes in Chinese. In addition, these euphemisms are based on the time slot and topic similarity for specific news hotspots and events. (2) Existing research used static word embeddings [12, 33] and bag-of-words model [36], which does not distinguish different meanings of the same word or capture the differences between semantically similar target keywords in the same category (e.g., cocaine and heroin in drug category). Therefore, we propose a new framework based on feature fusion and feature individualization modules, which can dynamically extract fusion features of sentences and words, obtain discriminative individual features of the target keywords, and effectively identify euphemisms in the corpus.

## 3 PROBLEM DESCRIPTION

Given sentences containing euphemisms $S$ and a set of target keywords $T$ as input: $s = [w_1, ..., w_i, euph, ..., w_m]$ (where $s \in S$, $euph$ is a euphemism), $T = \{t_1, ..., t_j, ..., t_n\}$. The task is to find the target keyword $t_j$ that refers to the euphemism $euph$. Taking the sentences in Table 1 for example, we aim to determine that the value for euphemism "nine" is the target keyword "gun" and "coke" refers to "cocaine".

## 4 FEATURE FUSION AND INDIVIDUALIZATION

We convert the euphemism identification task into a cloze task [28], masking out the euphemism in the sentence and finding the target keyword that best matches the mask. As illustrated in Figure 1, "[MASK]" refers to the blank of the sentence, and the goal is to find the best candidate for the blank, formalized as shown in Formula (1):

$$t^* = \arg\max P(t_j|s) \quad s \in S, j \in 1, ..., n \tag{1}$$

where $S$ is the set of sentences that masked out the euphemisms, $t_j$ is the j-th target keyword in the candidate set of a specific category (Drug, Weapon, or Sexuality), and $t^*$ is the target keyword that maximizes the probability of the blank given the masked sentence $s$.

Inspired by Zhu et al. [36], we use self-supervised learning to overcome the limitation of the lack of labeled datasets with accurate

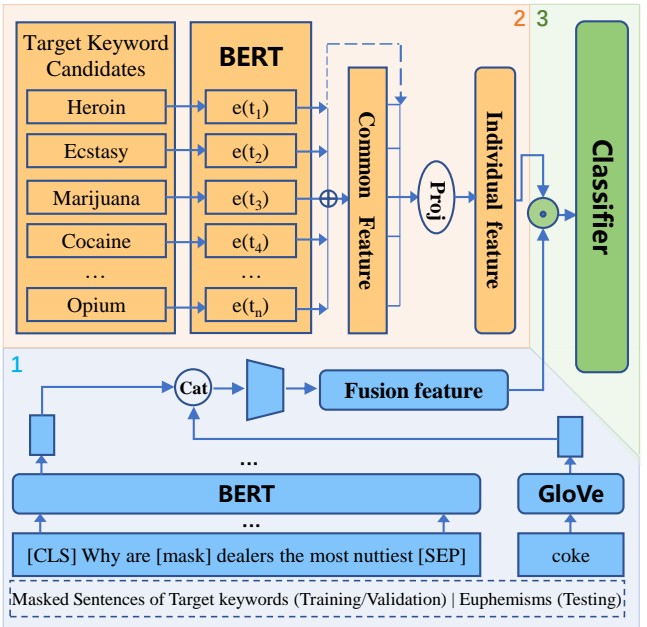

**Figure 2: Model framework.**

mapping relationships between euphemisms and the target keywords. In the training and validation phase, we take the sentences masking the target keywords (e.g., cocaine and heroin) as training samples, using the corresponding target keywords as labels for training. During the testing phase, we feed the sentences with the euphemisms masked into the trained model and finally specify the masked euphemism into the corresponding target keyword.

### 4.1 The Overall Framework

We convert the euphemism identification task into a cloze task and propose a feature fusion and individualization method (FFI) to recognize euphemisms. The framework of our proposed FFI is shown in Figure 2, which consists of three parts, namely 1) a feature fusion module, 2) a feature individualization module, and 3) a prediction module.

The feature fusion module extracts fusion features of masked sentences and words by integrating both dynamic global and static local information to enhance the semantics and make the features more discriminative. Meanwhile, the feature individualization module extracts discriminative individual features for target keywords to strengthen the difference between the features, thus addressing the challenge of subtle semantic differences between target keywords. Finally, the prediction module combines the aforementioned

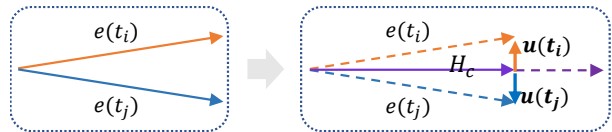

**Figure 3: Orthogonal decomposition process of target keyword vectors. The target keyword vectors $e(t_i)$ and $e(t_j)$ are orthogonally projected with their common feature vector $H_c$, to obtain the individual feature vectors $u(t_i)$ and $u(t_j)$ of the target keywords.**

fusion features and individual features to obtain the final features, using a classifier to complete the cloze task.

## 4.2 Feature Fusion Module

Euphemisms are usually identified according to their context, however if only context information is considered, similar contexts of distinct euphemisms may mislead the model to learn confusing information, leading to erroneous identification. For instance, the sentence, "We had already paid $70 for some shitty weed from a taxi driver but we were interested in some coke and the cubans", contains both euphemisms "weed" and "coke". It is difficult to distinguish between "weed" and "coke" if only sentence-level context information is considered.

According to previous research on the evolution of euphemisms [13] and common sense, the literal meaning of euphemisms is related to their secret meanings. For example, the literal meaning of weed and its true meaning of marijuana are both plants. Coke is a euphemism for cocaine, because the original coke is a drink containing cocaine. Notably, the euphemisms are easily distinguished in literal sense, such as weed and coke. Inspired by these, we introduce static local features of euphemisms representing their literal meanings into the dynamic global features of euphemism context, thus obtaining semantically rich and discriminative fusion features.

Due to Bert's success in dynamically extracting contextual features [6] and GloVe [22] in extracting static features [16], we use Bert model pre-trained on euphemism corpus to extract dynamic sentence features and GloVe to extract static word features. Take the sentence $s = [w_1, ..., w_i, [\text{MASK}], ..., w_m]$ ($s \in S$, where $S$ is the set of masked sentences) with the euphemism masked as the input of BERT model. $w_i$ refers to a token, and the special tokens "[CLS]" and "[SEP]" are boundary markers used to guide and end the input sequence. $w_{mask}$ refers to the original masked word. As shown in formula (2) and (3), $h_g(s) \in R^{d_g}$ and $h_l(w_{mask}) \in R^{d_l}$.

$$h_g(s) = \text{CLS\_BERT}([\text{CLS}]+w_1+...+w_i+[\text{MASK}]+w_m+[\text{SEP}]) \quad (2)$$

$$h_l(w_{mask}) = \text{GloVe}(w_{mask}) \quad (3)$$

Then, a simple but effective method is applied to fusion the dynamic global and static local features, as follows:

$$H(s) = \omega_f \cdot (h_g(s); h_l(w_{mask})) + b_f \quad (4)$$

where $\omega_f \in R^{d_g \times (d_g+d_l)}$, $b_f \in R^{d_g}$ are the model parameters, and (;) means concatenation.

**Table 3: Overview of the datasets. Num means categories of target keywords. Key_Entries means entries containing the target keywords.**
d/w/s_D1 = Drug/Weapon/Sexuality. d_D2 = Drug+Weapon, w_D2 = Weapon+Sexuality, s_D2 = Sexuality+Drug.
d/w/s_D3 = Drug+Sexuality+Weapon.

| Datasets | Entries | Num | Key_Entries |
|---|---|---|---|
| Drug | 1271907 | 33 | drug:4566 |
| Weapon | 3108988 | 9 | weapon:19003 |
| Sexuality | 2894869 | 12 | sexuality:7215 |
| d_D1 | 1271907 | 33 | drug:4566 |
| d_D2 | 4380895 | 33 | drug:7015 |
| d_D3 | 7275764 | 33 | drug:9570 |
| w_D1 | 3108988 | 9 | weapon:19003 |
| w_D2 | 6003857 | 9 | weapon:28923 |
| w_D3 | 7275764 | 9 | weapon:29046 |
| s_D1 | 2894869 | 12 | sexuality:7215 |
| s_D2 | 4166766 | 12 | sexuality:7288 |
| s_D3 | 7275764 | 12 | sexuality:30534 |

## 4.3 Feature Individualization Module

Motivated by Qin et al. [23], we hypothesize that there is a common semantic space between semantically similar target keywords to display their common features and that each target keyword has an independent space orthogonal to this space to display their individual discriminative features. Thus, to alleviate the problem of subtle semantic differences among target keywords, we employ a feature individualization module that projects target keyword features into a purified semantic space orthogonal to the common semantic space to get individual features for identification.

The projection decomposition process is depicted in Figure 3, and is explained using two-dimensional feature vectors of two target keywords for clarity. As observed from Figure 3, when the target keyword vectors $e(t_i)$ and $e(t_j)$ are proximate, distinguishing between the two becomes arduous. By utilizing the orthogonal projection method, we can strip off the common parts of the vectors and obtain the individual parts of each vector, denoted by $u(t_i)$ and $u(t_j)$, thereby augmenting the dissimilarity between them, making it easier to distinguish them.

Specifically, we encode each target keyword through the pre-trained Bert to obtain $e(t_j)$ and take their average encoding as the common encoding $H_c$. Then, we calculate the projection of each target keyword feature vector $e(t_j)$ onto the orthogonal direction of the common feature vector $H_c$ and finally obtain the discriminative individual feature $u(t_j)$.

$$e(t_j) = \text{CLS\_BERT}([\text{CLS}] + t_j + [\text{SEP}]) \quad j \in 1, ..., n \quad (5)$$

$$H_c = \frac{1}{n} \sum_{j=1}^{n} e(t_j) \quad (6)$$

$$\text{Proj}(a, b) = \frac{a \cdot b}{|b|} \frac{b}{|b|} \quad (7)$$

$$u(t_j) = \text{Proj}(e(t_j), (e(t_j) - H_c)) \quad j \in 1, ..., n \quad (8)$$

## 4.4 Prediction Module

After obtaining the fusion feature $H(s)$ and individual feature $u(t_j)$, the cloze task is finally achieved through the classifier. The probability of obtaining the selected target keyword for a given mask sentence is calculated by a combination of the fusion feature and individual feature:

$$P(t_j|s) = \frac{exp(\omega \cdot (u(t_j) \odot H(s)) + b)}{\sum_{j=1}^{n} exp(\omega \cdot (u(t_j) \odot H(s)) + b)} \quad (9)$$

where $\omega \in R^{d_g}$, $b \in R$ are the model parameters and $\odot$ is the element-wise multiplication. The objective of the training is to minimize the cross entropy between predicted results and true values:

$$loss = -\sum_{j=1}^{n} H_g log P(t_j|s) \quad (10)$$

where n is the number of target keyword subcategories in a specific category. In drug, weapon, or sexuality category, target keywords in the same subcategory hold identical meanings. $H_g$ is the one-hot vector of the ground truth.

## 5 EXPERIMENT

In this section, we evaluate the performance of FFI on the benchmark datasets presented by Zhu et al. [36] and a combination of these datasets and compare it with a set of baseline models.

### 5.1 Experimental Setup

*5.1.1 Datasets.* We empirically validated our proposed model on three separate datasets: Drug, Weapon, and sexuality [36]. In addition, we added a mixture of these datasets to verify our model generalization capability. These datasets are sourced from the Reddit website[3], Gab social networking services[4], Online Slang Dictionary[5], etc.

An overview of the datasets is presented in Table 3. d_D1, d_D2, and d_D3 respectively denote Drug, the mixed set of Drug and Weapon, and the mixed set of the three datasets, in which we aim to identify euphemisms for drug target keywords in these datasets. Similarly, to identify weapon euphemisms in w_D1, w_D2, w_D3, and sexuality euphemisms in s_D1, s_D2, s_D3. There are 33, 9, and 12 subcategories of target keywords corresponding to datasets Drug, Weapon, and Sexuality, respectively.

When training the model, the training and validation data must mask out the target keywords. When testing, the test data must mask out the euphemisms. Therefore, we require two kinds of inputs: 1) sentences from the original text corpus that mask out the target keywords (for training/validation) and sentences that mask out the euphemisms (for testing), and 2) a list of target keywords (e.g., heroin, cocaine, etc.). To evaluate our results, we need to rely on a ground truth list [36] of euphemisms and the corresponding target keywords, which should contain a one-to-one mapping from each euphemism to its true meaning. The ground truth list on Drug was compiled by the U.S. Drug Enforcement Administrator to provide a practical reference for law enforcement personnel [1].

The ground truth list on Weapon was sourced from the Online Slang Dictionary[6] and the Urban Thesaurus[7]. The ground truth list on Sexuality came from the Online Slang Dictionary. Due to the rapid evolution of the language used on social networks, it cannot be comprehensive or error-free, but it is the most reliable ground truth we can get.

Note that the ground truth list does not participate in the whole training process, but is only used to help evaluate the accuracy of euphemism identification, and no additional resources or supervision are required throughout the training process.

*5.1.2 Training details.* To exclude other factors from affecting the comparison with the baselines, we also trained the models separately on each dataset and split the training set and validation set in an 8:2 ratio of sentences that mask out the target keywords, while the test set comprised all sentences that mask out the euphemisms. Firstly, we pre-trained a Bert model based on bert-base-uncased[8] for MLM task only to extract dynamic global features (768 dimensions) of masked sentences and used a pretrained GloVe model to extract static local features (100 dimensions) of words. Then, we fine-tuned the model for the euphemism identification task. During pre-training, the maximum length of input sequence was set as 512, the batch size as 64, and the number of iterations as 3. For model training, the maximum length of input sequence was 128, and the batch size was 32. The initial learning rate was 5e-5, the warm-up step was 1000, and the optimizer AdamW [18] is based on a warm-up linear schedule. All experiments were conducted on a Linux server of Ubuntu 18.0.4 LTS version with a Tesla-V100 32G GPU.

*5.1.3 Evaluation Metrics.* Similar euphemisms refer to the target keywords with similar semantics makes it difficult to locate the target keyword of the euphemism precisely. For each euphemism, we generate a probability distribution for all target keywords. Given the nature of the output, we evaluate the top-K accuracy (Acc@k) [30], which measures how often the actual labels appear in the first k values of the ranked list we generate. To be consistent with the baselines, we also take the results of Acc@1, Acc@2, and Acc@3 for comparison.

### 5.2 Results and Analysis

We use four baselines to compare with FFI. These baselines include the method proposed by Zhu et al. [36] (the SOTA model, denoted as "SelfEDI"), the Word2vec baseline they created, and the other two baselines established by us.

- **Word2vec**: Use Word2vec to obtain word embeddings of all words, using cosine similarity to select the closest target keyword.
- **SelfEDI**: Use a bag-of-words model to extract sentence features at the sentence level, and train a multinomial logistic regression classifier to recognize euphemisms.
- **BERT_pre**: Use the pre-trained model obtained from a category corpus (such as Drug dataset) to extract the sentence

---

[3]https://www.reddit.com/
[4]https://gab.com/
[5]https://slangpedia.org/

[6]http://onlineslangdictionary.com/
[7]http://urbanthesaurus.org/
[8]https://huggingface.co/bert-base-uncased/

Table 4: Experimental results of baselines and the proposed FFI.

| Model | Drug | | | Weapon | | | Sexuality | | |
|---|---|---|---|---|---|---|---|---|---|
| | Acc@1 | Acc@2 | Acc@3 | Acc@1 | Acc@2 | Acc@3 | Acc@1 | Acc@2 | Acc@3 |
| Word2vec | 0.07 | 0.14 | 0.21 | 0.10 | 0.27 | 0.40 | 0.17 | 0.22 | 0.42 |
| SelfEDI | 0.20 | 0.31 | 0.38 | 0.33 | 0.51 | 0.67 | 0.32 | 0.55 | 0.64 |
| BERT_pre | 0.21 | 0.26 | 0.33 | 0.35 | 0.54 | 0.70 | 0.36 | 0.55 | 0.64 |
| BERT_ft | 0.25 | 0.33 | 0.40 | 0.40 | 0.59 | 0.72 | 0.38 | 0.55 | 0.65 |
| FFI_fusion | 0.27 | 0.36 | 0.38 | 0.40 | 0.59 | 0.67 | 0.39 | 0.52 | 0.62 |
| FFI_diff | 0.29 | 0.37 | 0.40 | 0.42 | 0.64 | 0.72 | 0.42 | 0.55 | 0.65 |
| **FFI** | **0.31** | **0.39** | **0.43** | **0.42** | **0.65** | **0.72** | **0.46** | **0.57** | **0.67** |

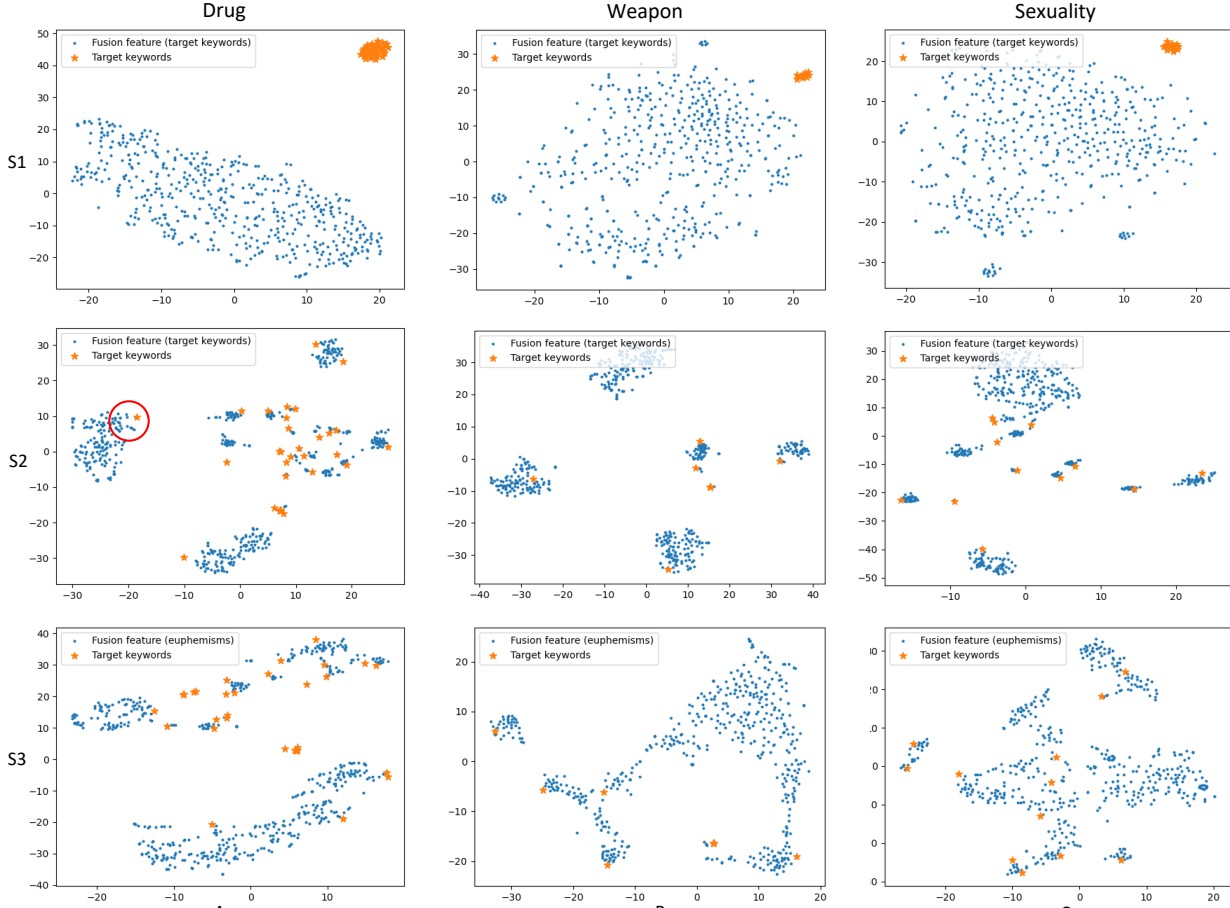

Figure 4: A/B/C.Visualization of FFI on Drug/Weapon/Sexuality Datasets. S1 represents the stage before training, while S2 and S3 represent the stages after training. The orange stars denote the target keyword features, 33 in drug category, 9 in weapon category, and 12 in sexuality category. The blue dots indicate the fusion features with the target keywords masked in S1 and S2, and the fusion features with the euphemisms masked in S3, 512 randomly taken in each category of drug, weapon or sexuality. The red circle in stage S2 of subplot A masks that the fusion features are clustered with the target keyword feature.

features (fixed parameters), and train a multinomial logistic regression classifier to recognize euphemisms.

- **BERT_ft**: Use the pre-trained model obtained on a specific corpus to extract the sentence features (updatable parameters), then perform element-wise multiplicative with the

feature vectors, and finally do the fine-tuning to complete the euphemism identification cloze.

---

[9]https://replicate.com/stability-ai/stablelm-tuned-alpha-7b
[10]https://modelscope.cn/studios/damo/mPLUG-Owl/summary
[11]https://huggingface.co/models?other=llama-2
[12]https://platform.openai.com/docs/api-reference/introduction

**Table 5: Experimental results of FFI against the LLMs. "Cost/S" represents the average time and cost per sentence. "-" means that the models refuse to answer such questions involving inappropriate content.**

| Model | Drug | Weapon | Sexuality | Cost/S |
|-------|------|--------|-----------|--------|
| StableLM[9] | 0.02 | 0.03 | 0.12 | 2.08S/0.00475$ |
| mPLUG-Owl[10] | 0.02 | 0.13 | 0.15 | 2.35S/0.00541$ |
| Llama2[11] | 0.17 | - | - | 18.23S/0.05833$ |
| GPT3.5[12] | **0.33** | 0.17 | 0.42 | 1.12S/0.00035$ |
| FFI | 0.31 | **0.42** | **0.46** | **0.27S/0.00003**$ |

*5.2.1 Comparison with baselines.* Table 4 summarizes the euphemism identification results (the top two rows are taken directly from Zhu et al. [36]). To be fair, the results of all models are taken from the parameters that make the results the best. Our FFI achieves the best performance. Specifically, FFI outperforms the SOTA model (Self-EDI) by 11%, 9%, and 14% in top1 accuracy value on three datasets, respectively.

Word2vec showed poor performance as it did not capture the subtle differences among the target keywords. Compared to Word2vec, BERT_pre, and SelfEDI showed better performance, both extracting sentence semantic information relatively well. The results of BERT_pre on three datasets are slightly better than those of SelfEDI. Compared to SelfEDI (using a bag-of-words model to extract sentence features), BERT_pre uses BERT encoding, which preserves the sequential order of words and considers the semantic connections between words, resulting in obtaining sentence features with richer semantics. The results of BERT_ft are better than those of BERT_pre. The latter is based on a feature approach, using BERT as a feature extractor, while the former uses a fine-tuning approach, which is clearly superior to the feature-based approach.

Experimental results suggest that the exploration of alleviating the problem of subtle semantic differences among euphemisms referred to target keywords with similar semantics can maximally exploit the discriminative features of the euphemisms and target keywords, so as to improve the identification performance.

*5.2.2 Comparison with LLMs.* With recent advances in large language models, series of tasks in natural language processing have been solved well [34]. To investigate the effectiveness of the LLMs in euphemism identification, we directly use the current best LLMs to identify euphemisms, and the results obtained are shown in Table 5. We observe: 1) Our FFI model beats almost all the LLMs; 2) In the four LLMs, GPT3.5 is the best and most stable for euphemism identification; 3) Compared with our FFI , the time-consuming of the LLMs is about 3-8 times that of ours, and the cost is about 10-200 times that of our FFI.

Although the recognition accuracy of GPT3.5 on the Drug dataset is 2 percentage points higher than that of our FFI, there is still a lack of understanding of rare euphemisms. We observe that GPT3.5 has a recognition accuracy rate of 100% for common euphemisms, such as "weed", "pot" and "coke", while its recognition accuracy for rare euphemisms, such as "ice", is almost 0. Among the correctly recognized euphemism, the proportion of common euphemism is 57.7%, providing a significant positive impact on the recognition result. Thus, we deduce that GPT3.5 recognizes euphemisms by

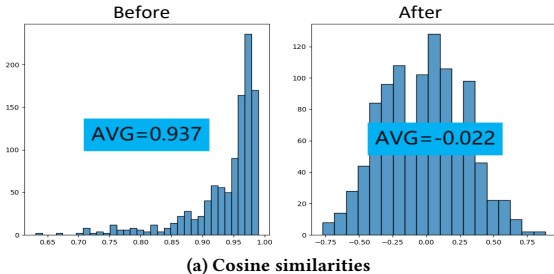

**(a) Cosine similarities**

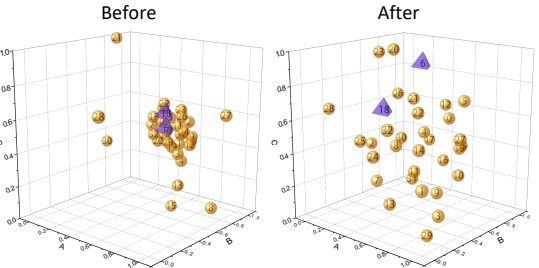

**(b) 3D representation distribution**

**Figure 5: (a) Cosine similarities and (b) 3D representation distribution of the target keywords in the drug category before and after the orthogonal projection is conducted. In (a), the abscissa represents the cosine similarity values, and the ordinate represents the corresponding number to abscissa. In (b), the purple tetrahedron No.6 and No.18 refers to "cocaine" and "marijuana" respectively.**

common sense based on a large amount of corpus, rather than by the context of euphemisms, as detailed in the supplementary material. Based on the above analysis, our FFI model is better than the LLMs in terms of identification results, time, cost, etc.

*5.2.3 Visualization.* To further substantiate the soundness of the FFI, We map the distributions of context and target keyword features to a two-dimensional coordinate space by t-SNE, as shown in Figure 4. After using our FFI, across the three datasets, we observed: 1) both the fusion features and target keyword features become more dispersed and easier to distinguish, and the distance between them decreases; 2) the fusion features tend to converge on specific target keywords. These show that our method can differentiate euphemisms with similar contexts and target keywords with similar semantics, while preserving the euphemism context to match the corresponding target keywords, further proving the effectiveness of our method.

## 5.3 Ablation Studies

To investigate the efficacy of each component of FFI, We conducted experiments using or removing the feature fusion or feature individualization module on the three datasets. BERT_ft is the base version of FFI, using only the BERT backbone to fine-tune the euphemism identification task. FFI_fusion signifies that feature fusion is added on the basis of BERT_ft, while FFI_diff indicates that feature individualization is added on the basis of BERT_ft.

Experimental results are presented in Table 4. It can be seen that the FFI_fusion or FFI_diff always obtains a larger improvement

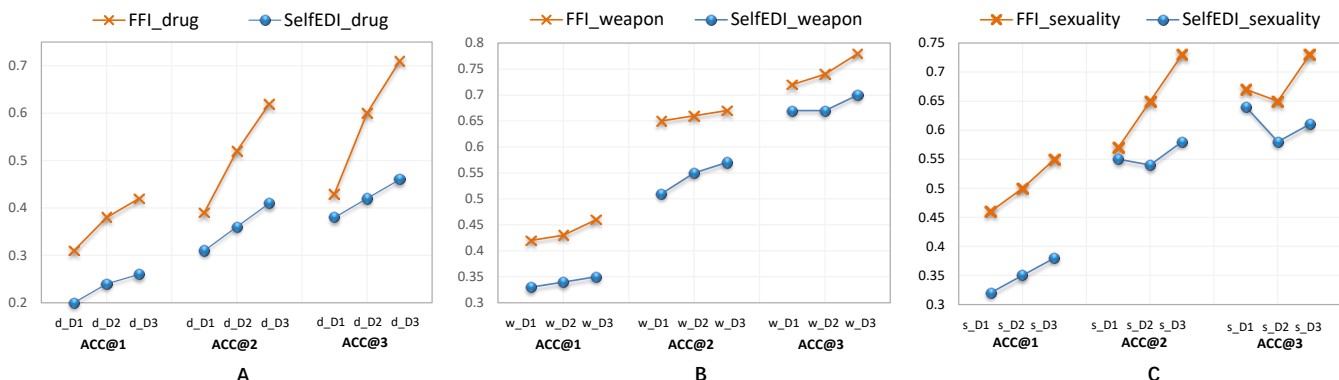

**Figure 6: A.Identification results of drug euphemisms on d_D1, d_D2, and d_D3. B.Identification results of weapon euphemisms on w_D1, w_D2, and w_D3. C.Identification results of weapon euphemisms on s_D1, s_D2, and s_D3.**

than BERT_ft. This is a strong suggestion on the promotional effect of feature fusion or feature individualization in maximizing the discrimination of the euphemisms or target keywords. However, the feature individualization contributes more to the improvement of top1 identification accuracy than the feature fusion does, with an increase of 2-3%. The effects of feature fusion and feature individualization on the model are analyzed in detail below.

*5.3.1 Feature Fusion.* As can be seen from Table 4, utilizing the feature fusion module to obtain fusion features of masked sentences and words has improved the model's performance, particularly in terms of the top1 accuracy. If only sentence-level or word-level semantic information is used, it is difficult to distinguish the euphemisms with similar contexts. Taking sentence 1 in Table 1 for example, we get two euphemism sentences: $s_1$ ("We had already paid \$70 for some shitty [mask] from a taxi driver but we were interested in some coke and the cubans."), $s_2$ ("We had already paid \$70 for some shitty weed from a taxi driver but we were interested in some [mask] and the cubans."). The similarity between $s_1$ and $s_2$ is 0.9903 before fusion is conducted, and drops to 0.9310 after fusion, which proves the efficacy of the feature fusion module.

*5.3.2 Feature Individualization.* Table 4 shows that the orthogonal projection of feature individualization has significantly improved the model's performance. From the intuitive data, the average cosine similarity between target keywords before projection is 0.937, and it drops to -0.022 after projection, as shown in Figure 5a, where the similarity matrix is shown in Appendix Table 7. Further 3D representation distribution is shown in Figure 5b. It is apparent that the dispersion of target keywords in the drug category is more distinct after conducting orthogonal projection. Taking "marijuana" and "cocaine" as examples, the cosine similarity value between "marijuana" and "cocaine" is 0.98404 before projection, but decreases to 0.27118 after projection. Above all, the orthogonal projection of feature individualization can alleviate the problem of subtle semantic differences among target keywords.

### 5.4 Generalization of FFI

We augmented the training data by amalgamating the Drug, Weapon, and Sexuality datasets to assess the generalization of the model. As shown in Table 3, d/w/s_D1, d/w/s_D2, and d/w/s_D3 comprise a growing number of target keywords, implying a gradual increase in the size of the training data. Figure 6 shows the top1, top2, and top3 results of identifying euphemisms related to drug, weapon, and sexuality on the d/w/s_D1, d/w/s_D2, and d/w/s_D3 datasets, respectively.

In Figure 6, the result curve obtained by FFI clearly lies above that of SelfEDI. Additionally, the line graph presents an overall upward trend for both FFI and SelfEDI in the identification task of the three categories as the training data gradually increases (from D1_d/w/s, D2_d/w/s to D3_d/w/s), with the top1 recognition rate showing an absolute upward trend. However, the performance increase achieved by FFI is 83-250% higher than that of SelfEDI, indicating that our method exhibits a faster performance improvement as the training data increases.

When identifying the sexuality euphemisms, the top2, and top3 results did not exhibit a consistent upward trend. Upon analysis, the sexuality data are extremely unbalanced, with the "sex" subcategory accounting for a whopping 72% of the training dataset, while the remaining subcategories only account for 2.5%. Despite this imbalance, the top1 accuracy was still as expected and continued to improve as the amount of training data increased.

Overall, the model demonstrates commendable performance in terms of generalization.

## 6 CONCLUSION

We formulate the euphemism identification task as a cloze task and propose a FFI method for euphemism identification. In FFI, a feature fusion module is employed to capture both dynamic global and static local features to enhance discrimination among different euphemisms in similar contexts. Jointly, a feature individualization module is used to project features into the orthogonal space, extracting distinct individual features of each target keyword. Therefore, FFI can effectively differentiate euphemisms in similar contexts referring to target keywords with similar semantics. Extensive experiments demonstrate the feasibility and state-of-the-art performance of our FFI. Furthermore, the performance of FFI improves rapidly as the amount of training data increases, indicating its strong generalization ability.

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

# A   ETHICAL STATEMENT AND LIMITATIONS

## A.1   Ethical Statement.

This study heavily drew upon user-generated content, and the dataset was legally obtained following the guidelines set forth by Zhu et al. [36]. and adhered to strict privacy standards, ensuring the absence of any personally identifiable information such as real names, email addresses, IP addresses, etc. Moreover, the data comprises information from the year 2018 or earlier, which significantly reduces any sensitivity they may have.

In the context of previous research conducted on online anonymity marketplaces [4, 26], Carnegie Mellon University's Institutional Review Board (IRB) explicitly stated that analyses based on user-generated content do not constitute human research and thus do not fall under the jurisdiction of the IRB, provided that (1) the data analyzed are publicly posted and not a result of direct interaction between the researcher and the poster, (2) no private identifiers or personally identifiable information are associated with the data, and (3) the study did not correlate various public data sources to infer private information. Our studies fully adhere to all the above conditions.

## A.2   Limitations.

Although our euphemism identification method achieves the best performance at present, there are some limitations.

1) Promotion of multimodal data: Our existing method only considers textual data, thereby neglecting the inclusion of other

modalities prevalent in social media, such as images, videos, or audio. In social media or underground marketplaces, text frequently intertwines with diverse modal data. In the future, we will consider integrating multi-modal data processing to facilitate information extraction and thus identify euphemisms more effectively.

2) Gap from the training set to test set: Since there is no labeled dataset for training the euphemism identification problem. During the training phase, sentences containing the target keywords are used with the target keywords masked out, while the corresponding target keywords serve as labels. Nevertheless, during testing, sentences containing euphemisms are used, with the euphemisms masked out. As a result, the training and test data diverge in terms of their distribution resulting in a relatively large gap between them, which can be seen from 4. Although our model has achieved the current best performance and uses feature fusion aiming to bridge the gap between training and test data, there remains scope for further improvement, and this will be the focus of our subsequent research.

## B SIMILARITY METRIC

The feature individualization module employs the orthogonal decomposition to project the target keywords originally in a dense space into a sparse space, which leads to a notable reduction in the similarity values among the target keywords, rendering them more distinguishable. Table 7 shows part of the similarity values between target keywords before and after the orthogonal projection. It can be seen that all the similarity values in the table become smaller after orthogonal projection, further substantiating the effectiveness of our method.

## C LLMS FOR EUPHEMISM IDENTIFICATION

In this paper, we compared our proposed FFI model to four current best large language models (LLMs) for euphemism identification task, namely, GPT-3.5-turbo(GPT3.5 for short), Llama2, mPLUG-Owl, and StableLM. These LLMs are described in detail below.

### C.1 Introduction of LLMs

We briefly introduce the four LLMs from the model type, parameter number, maximum text input length, cost and other aspects, as shown in Table 8. GPT3.5 and stableLM are both natural language processing models, while Llama2 and mPlug-Owl are multimodal processing models which are more expensive. For details and interfaces about the LLMs, see the footnote link address.

### C.2 Result Analysis

When using the GPT3.5 and StableLM interfaces to identify euphemisms, we used four content templates, as shown in Table 9. From Table 9, we observe that the results varies according to the content templates, and GPT3.5 is relatively stable compared to StableLM. However, the results are not consistent across different models and different datasets, indicating the randomness of the output results of these large language models. For the other multi-modal processing models, i.e., Llama2 and mPlug-Owl, which are too expensive to use four templates for testing, so that we only use Template 1 to test the identification accuracy. Finally, we take the best result on each dataset and record it to Table 4 in the body part.

It's obvious that GPT3.5 performs the best among the four LLMs, and outperforms our proposed FFI by two percentage points on the Drug dataset. When using Llama2 API interface or web UI to test the identification accuracy on Weapon or Sexualtiy dataset, it informs that it is inappropriate to discuss such a topic and refuses to answer questions, while we can only test on the Drug dataset via web UI. We present case studies of the LLMs in the following sections.

### C.3 Case studies

Through the analysis of the euphemism identification results of the four LLMs, we have the following two findings:

1) GPT3.5 performs the best among the four LLMs with strong understanding of euphemisms. However, it still lacks understanding of the relatively rare euphemisms, and the recognition rate of commonly used euphemisms is almost 100%. As shown in Figure 7a, it can always idenfy "weed" (a common euphemism) to its true meaning "marijuana", while having no idea of "ice" in euphemistic use of "methamphetamine" (Figure 7b).

2) GPT3.5 is relatively stable as the identification results of common euphemisms are correct while the other LLMs are not. That's why the other LLMs perform far worse than GPT3.5 in euphemism identification. Take the mPLUG-Owl model for example, when we ask it for the same question about the meaning of the euphemism in the sentence four times, it gives completely different answers, as shown in Figure 8.

**Table 6: Part of the target keywords in the drug category.**

| ID | Target Keywords |
|----|-----------------|
| 0 | acetaminophen and oxycodone combination |
| 1 | Alprazolam |
| 2 | amphetamine |
| 3 | amphetamine and dextroamphetamine combination |
| 4 | buprenorphine and naloxone combination |
| 5 | clonazepam |
| 6 | cocaine |
| 7 | crack cocaine |
| 8 | ecstasy |
| 9 | fentanyl |
| 10 | flunitrazepam |
| 11 | gamma-hydroxybutyric acid |
| 12 | heroin |
| 13 | hydrocodone |
| 14 | hydromorphone |
| 15 | ketamine |
| 16 | khat |
| 17 | lysergic acid diethylamide |
| 18 | marijuana |
| 19 | ... |

**Table 7: The cosine similarity values between part of the target keywords in the drug category before and after orthogonal projection was conducted. No.6 and No.18 refers to "cocaine" and "marijuana" respectively**

| Before | 0 | 1 | 2 | 3 | 4 | 5 | 6 | ... |
|---|---|---|---|---|---|---|---|---|
| 0 | 1 | 0.93579 | 0.95589 | 0.88444 | 0.97922 | 0.9776 | 0.98076 | |
| 1 | 0.93579 | 1 | 0.92202 | 0.8813 | 0.92497 | 0.9166 | 0.93248 | |
| 2 | 0.95589 | 0.92202 | 1 | 0.9534 | 0.94525 | 0.949 | 0.95726 | |
| 3 | 0.88444 | 0.8813 | 0.9534 | 1 | 0.86627 | 0.88073 | 0.89448 | |
| 4 | 0.97922 | 0.92497 | 0.94525 | 0.86627 | 1 | 0.9805 | 0.96735 | |
| 5 | 0.9776 | 0.9166 | 0.949 | 0.88073 | 0.9805 | 1 | 0.9805 | |
| 6 | 0.98076 | 0.93248 | 0.95726 | 0.89448 | 0.96735 | 0.9805 | 1 | |
| 7 | 0.98765 | 0.93848 | 0.95193 | 0.87878 | 0.97608 | 0.97738 | 0.98511 | |
| 8 | 0.97875 | 0.93468 | 0.94151 | 0.86306 | 0.98797 | 0.97902 | 0.96776 | |
| 9 | 0.97974 | 0.91653 | 0.94617 | 0.86884 | 0.98273 | 0.98362 | 0.97401 | ... |
| 10 | 0.9358 | 0.88706 | 0.95108 | 0.90685 | 0.95347 | 0.95249 | 0.93447 | |
| 11 | 0.98201 | 0.92359 | 0.9553 | 0.88771 | 0.98497 | 0.98798 | 0.9788 | |
| 12 | 0.97829 | 0.92588 | 0.92293 | 0.84195 | 0.96305 | 0.96913 | 0.97594 | |
| 13 | 0.97452 | 0.929 | 0.95682 | 0.90218 | 0.98179 | 0.97925 | 0.96716 | |
| 14 | 0.96879 | 0.89648 | 0.94124 | 0.8636 | 0.97425 | 0.98618 | 0.96475 | |
| 15 | 0.90878 | 0.89057 | 0.87473 | 0.80518 | 0.9192 | 0.92579 | 0.91216 | |
| 16 | 0.97743 | 0.92129 | 0.94097 | 0.86879 | 0.97689 | 0.98363 | 0.97902 | |
| 17 | 0.97467 | 0.91505 | 0.93579 | 0.85561 | 0.98715 | 0.98306 | 0.96582 | |
| 18 | 0.97993 | 0.94131 | 0.95373 | 0.89399 | 0.96473 | 0.97504 | **0.98404** | |

| After | 0 | 1 | 2 | 3 | 4 | 5 | 6 | ... |
|---|---|---|---|---|---|---|---|---|
| 0 | 1 | 0.12843 | -0.01018 | -0.18083 | 0.22706 | -0.12476 | 0.15831 | |
| 1 | 0.12843 | 1 | 0.14171 | 0.19981 | -0.01622 | -0.36394 | 0.05465 | |
| 2 | -0.01018 | 0.14171 | 1 | 0.71383 | -0.17843 | -0.3343 | 0.00869 | |
| 3 | -0.18083 | 0.19981 | 0.71383 | 1 | -0.36669 | -0.34613 | -0.0358 | |
| 4 | 0.22706 | -0.01622 | -0.17843 | -0.36669 | 1 | 0.16963 | -0.25071 | |
| 5 | -0.12476 | -0.36394 | -0.3343 | -0.34613 | 0.16963 | 1 | -0.01093 | |
| 6 | 0.15831 | 0.05465 | 0.00869 | -0.0358 | -0.25071 | -0.01093 | 1 | |
| 7 | 0.43728 | 0.16219 | -0.16536 | -0.31368 | 0.05256 | -0.23536 | 0.29956 | |
| 8 | 0.17377 | 0.13514 | -0.30029 | -0.44136 | 0.58551 | 0.05431 | -0.28916 | |
| 9 | 0.13658 | -0.24153 | -0.26348 | -0.42359 | 0.35807 | 0.17991 | -0.13951 | ... |
| 10 | -0.29124 | -0.15924 | 0.3156 | 0.30853 | 0.17083 | 0.01904 | -0.34663 | |
| 11 | 0.06264 | -0.23275 | -0.17149 | -0.24227 | 0.35319 | 0.229 | -0.14951 | |
| 12 | 0.38907 | 0.09866 | -0.42432 | -0.49374 | 0.01902 | -0.0006 | 0.30549 | |
| 13 | 0.01363 | 0.03731 | 0.06633 | 0.1157 | 0.3753 | 0.07143 | -0.30645 | |
| 14 | -0.06919 | -0.42671 | -0.20065 | -0.3464 | 0.20883 | 0.49628 | -0.23968 | |
| 15 | -0.12337 | 0.14972 | -0.23479 | -0.22328 | 0.09846 | 0.13653 | -0.07761 | |
| 16 | -0.04844 | -0.2029 | -0.48961 | -0.49469 | 0.07171 | 0.08907 | -0.005 | |
| 17 | 0.03185 | -0.19902 | -0.41871 | -0.53254 | 0.56413 | 0.26113 | -0.34295 | |
| 18 | 0.10998 | 0.2262 | -0.09469 | -0.04879 | -0.36916 | -0.31779 | **0.27118** | |

**Table 8: Introductions of the four LLMs. MM = Multi-Modal, T = Trillion, B = Billion.**

| LLMs | Type | Parameters | Maximum input | Cost | Institution |
|------|------|-----------|---------------|------|-------------|
| **GPT3.5** | NLP | 1.2T | 4096 tokens | 0.015$/1k tokens | OpenAI |
| **StableLM** | NLP | 3B - 7B | 4096 tokens | 0.0023$/second | Stability AI |
| **Llama2** | MM | 7B - 70B | 4096 tokens | 1.05$/hour | Meta |
| **mPLUG-Owl** | MM | 7B | 2048 tokens | 0.0023$/second | DAMO Academy |

**Table 9: Content templates and results.**

| Template | GPT3.5 | | | StableLM | | |
|----------|--------|--------|-----------|----------|--------|-----------|
| | Drug | Weapon | Sexuality | Drug | Weapon | Sexuality |
| 1. Among the following candidates:{T}, in sentence:{s}, euphemism:{euph} refers to which one. | 0.3314 | 0.1543 | 0.4041 | 0.0214 | 0.0149 | 0.1184 |
| 2. Among the following candidates:{T}, in sentence:{s}, {euph} refers to which one. | 0.2971 | 0.1686 | 0.3876 | 0.0160 | 0.0248 | 0.1061 |
| 3. The true meaning of euphemism:{euph} in "{s}" refers to which of the following candidates:{T}. | 0.3257 | 0.1564 | 0.4204 | 0.0053 | 0.0299 | 0.1102 |
| 4. The true meaning of "{euph}" in "{s}" refers to which of the following canditates:{T}. | 0.3029 | 0.1482 | 0.0106 | 0.0107 | 0.0199 | 0.1020 |

"T" refers to the fixed target keyword candidate list: {0:acetaminophen and oxycodone combination, 1:alprazolam, 2:amphetamine, 3:amphetamine and dextroamphetamine combination, ... }

"s" refers to the test sentence containing euphemisms.

"euph" refers to the euphemism in the test sentence that need to be identified.

In the sentence "We had already paid $70 for some shitty weed from a taxi driver but we were interested in some coke and the cubans.", what is the true meaning of "weed" ?

In the sentence you provided, the term "weed" is a colloquial and slang term that refers to marijuana or cannabis, which is a psychoactive drug derived from the Cannabis plant. The sentence suggests that the speaker had already paid $70 for poor-quality marijuana from a taxi driver but now they are interested in obtaining cocaine ("coke") and possibly interacting with individuals from Cuba ("the Cubans"). It's worth noting that drug use can have legal and health implications, and it is important to prioritize personal safety and adhere to the laws and regulations of your jurisdiction.

Among the following candidates,in "sentence": "loyal consistent and trustworthy weed dutchmagic dutchdreams mdma nelson muntz cocaine london underground", "euphemism": "weed" refers to which one, please give the index number. Candidates:{0:acetaminophen and oxycodone combination, 1:alprazolam, 2:amphetamine, 3:amphetamine and dextroamphetamine combination, 4:buprenorphine and naloxone combination, 5:clonazepam, 6:cocaine, 7:crack cocaine, 8:ecstasy, 9:fentanyl, 10:flunitrazepam, 11:gamma-hydroxybutyric acid, 12:heroin, 13:hydrocodone, 14:hydromorphone, 15:ketamine, 16:khat, 17:lysergic acid diethylamide, 18:marijuana, 19:marijuana concentrates hash oil, 20:mescaline,E2 21:methamphetamine, 22:methylphenidate ritalin concerta, 23:morphine, 24:opium, 25:oxaydo, 26:phencyclidine, 27:promethazine, 28:psilocybin mushrooms, 29:synthetic cannabinoids, 30:synthetic cathinones, 31:steroids, 32:u-47815}

In the given "sentence," the "euphemism" "weed" corresponds to:

18: marijuana

Among the following candidates,in "sentence": "loyal consistent and trustworthy weed dutchmagic dutchdreams mdma nelson muntz cocaine london underground", "euphemism": "weed" refers to which one, please give the index number. Candidates:{0:acetaminophen and oxycodone combination, 1:alprazolam, 2:amphetamine, 3:amphetamine and dextroamphetamine combination, 4:buprenorphine and naloxone combination, 5:clonazepam, 6:cocaine, 7:crack cocaine, 8:ecstasy, 9:fentanyl, 10:flunitrazepam, 11:gamma-hydroxybutyric acid, 12:heroin, 13:hydrocodone, 14:hydromorphone, 15:ketamine, 16:khat, 17:lysergic acid diethylamide, 18:marijuana, 19:marijuana concentrates hash oil, 20:mescaline,E2 21:methamphetamine, 22:methylphenidate ritalin concerta, 23:morphine, 24:opium, 25:oxaydo, 26:phencyclidine, 27:promethazine, 28:psilocybin mushrooms, 29:synthetic cannabinoids, 30:synthetic cathinones, 31:steroids, 32:u-47815}

**(a) Identification of "weed"**

In the sentence "For all vendors of ice, it seems pretty obvious that it is not as pure as they market it.", what is the true meaning of euhemism "ice" ?

In the context of the sentence you provided, the word "ice" is not used as a euphemism. It is used literally to refer to the frozen form of water. Euphemisms are typically used to replace harsh or taboo words or phrases with milder or more socially acceptable ones. However, in this sentence, "ice" does not have an alternative meaning or serve as a euphemism for something else. It is simply referring to the physical substance of frozen water.

In the sentence "For all vendors of ice, it seems pretty obvious that it is not as pure as they market it.", what is the true meaning of euphemism "ice" ?

I apologize for any confusion. If the sentence is using the term "ice" as a euphemism, it would require additional context or knowledge about the specific euphemism being referred to. Without further information, it is difficult to determine the exact euphemistic meaning of "ice" in that sentence.

**(b) Identification of "ice"**

Among the following candidates,in "sentence": "captain hymenwrecker went as far as to obtain custody of filipino children for purposes of sex and child porn production and thats just for, "euphemism": "porn" refers to which one, please give the index number. Candidates:{0:anus, 1:breast breasts nipple nipples, 2:genitals private parts, 3:perineum, 4:pornography, 5:masturbate, 6:sex, 7:condom condoms, 8:promiscuous, 9:semen, 10:lgbt, 11:prostitution}

In the given "sentence," the "euphemism" "porn" corresponds to:

4: pornography

Among the following candidates,in "sentence": "captain hymenwrecker went as far as to obtain custody of filipino children for purposes of sex and child porn production and thats just for, "euphemism": "porn" refers to which one, please give the index number. Candidates:{0:anus, 1:breast breasts nipple nipples, 2:genitals private parts, 3:perineum, 4:pornography, 5:masturbate, 6:sex, 7:condom condoms, 8:promiscuous, 9:semen, 10:lgbt, 11:prostitution}

In the given "sentence," the "euphemism" "porn" corresponds to:

4: pornography

**(c) Identification of "porn"**

**Figure 7: Cases of GPT3.5.**

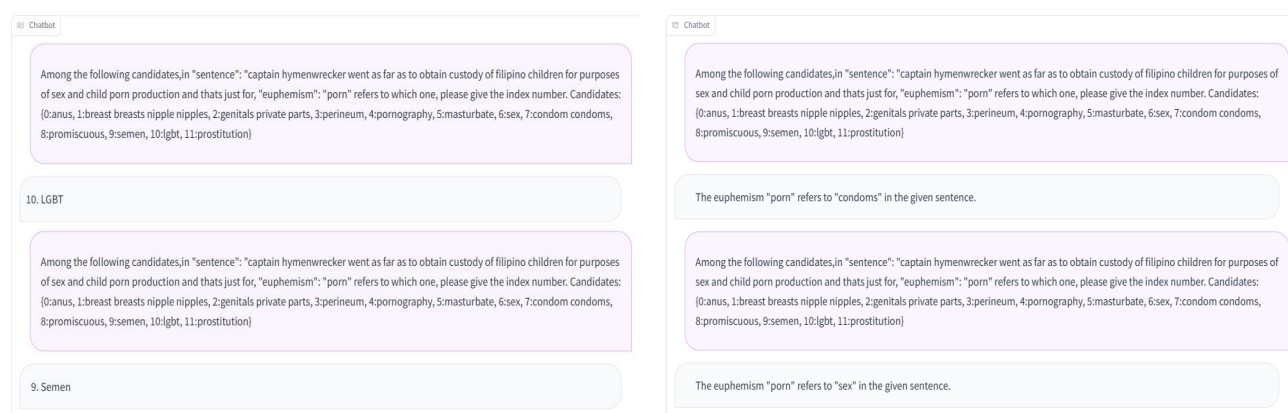

**Figure 8: Cases of mPLUG-Owl.**

