# OpenReview forum: "Euphemism Identification via Feature Fusion and Individualization"
_ACM.org/TheWebConf/2024/Conference — TheWebConf24 Oral_

### Official Review · Reviewer_FpzQ · 2023-11-22

**Novelty:** 5
**Technical Quality:** 5

**Review:**

Strengths:
1. The task and the problem is very interesting and important.
2. Authors perform different ablation studies which is good.
3. The paper is easy to follow.

Weaknesses:
1. Put a warning before the abstract to let the readers know that the paper contains some sensitive content.
2. Lines 92-99: I think these motivating goals can be explained more clear to highlight how your work is different.
3. The paper is about euphemism identification what about discovery of euphemism? I know that this is not the main goal of the paper but some suggestions on what can be used to first identify euphemism can be helpful.
4. Lines 745-799 where you discuss problems with GPT3.5, I am not fully convinced on the reasons brought by the authors. I think even their approach has this limitation. I think this is not a good justification. I would suggest authors to revisit this part of the paper and improve the justification.
5. Another issue was claiming their approach is not time consuming. I am assuming by time consuming they mean inference time, but it is worth nothing that their approach requires training of a separate model while these foundation models can be used for various applications, so again these justifications are not fully convincing to me. There is a trade-off on what the user can use for this particular application.

**Questions:**

Refer to some questions I have pointed out under weaknesses of the paper.

**Reviewer Confidence:**

3: The reviewer is confident but not certain that the evaluation is correct

**Scope:**

4: The work is relevant to the Web and to the track, and is of broad interest to the community

---

### Official Review · Reviewer_9KHd · 2023-11-22

**Novelty:** 7
**Technical Quality:** 6

**Review:**

This paper studies an important problem in social media analysis: how to recognize euphemisms. It insightfully make use of the language knowledge encoded in LLMs to detect the semantic similarity for euphemisms recognization. The idea is novel and the result is impressive. All the limitations that I can come up with are discussed by the authors in their appendix. But please include a warning ahead of the paper to warn the readers that there are sensitive contents in the paper.

**Questions:**

N/A

**Reviewer Confidence:**

3: The reviewer is confident but not certain that the evaluation is correct

**Scope:**

4: The work is relevant to the Web and to the track, and is of broad interest to the community

---

### Official Review · Reviewer_BEJJ · 2023-11-23

**Novelty:** 5
**Technical Quality:** 4

**Review:**

The paper aims to challenge the euphemism identification task, but with a main difference of formulating the euphemism identification task as a cloze task, and further propose a feature fusion and individualization (FFI) method for euphemism identification, which consists of three components: 1) a feature fusion model to extract fusion features of masked sentences and words by integrating both dynamic global and static local features to enhance discrimination among different euphemisms in similar contexts, 2) a feature individualization module to extract discriminative individual features for each target keyword by projecting features into the orthogonal space, and 3) a prediction module that combines the resulting fusion features and individual features into a classifier to obtain the final features for completing the cloze task. In this way, FFI can effectively differentiate euphemisms in similar contexts referring to target keywords with similar semantics. Experiments on three datasets of Drug, Weapon, and Sexuality and a mixture of these datasets validate the effectiveness and generalization of FFI, respectively.
Reason to accept:
1. This paper is well-written and easy to follow.
2. It is a good idea to convert euphemism identification into a cloze task to cope with the situation. There is no labeled dataset for training the euphemism identification problem.
3. The experiment is sufficient, and the analysis is reasonable.
Reason to reject:
1. There is an error in Eq. (2), the correct expression should be “hg(s)=CLS_BERT([CLS]+w1+…+wi+[MASK]+…+wm+[SEP])”.
2. The main novelty of the proposed method is to convert the euphemism identification task into a cloze task. However, the method is relatively simple and mainly relies on two techniques: feature concatenation and orthogonal projection.
3. The four baselines covered in the paper are relatively simple and not very competitive.
4. As far as the experimental results are concerned, only the improvement in Acc@1 is significant. In contrast, the improvement in Acc@2 and Acc@3 is fragile, and the authors did not analyze the potential reasons for this.
5. Lack of reproducible content, including data and code.
6. In Section 5.1.2, what is the rationale behind the authors pretraining a BERT base with only the MLM objective instead of directly employing the out-of-the-box one with MLM and NSP objectives?

**Questions:**

See Reason to reject.

**Reviewer Confidence:**

3: The reviewer is confident but not certain that the evaluation is correct

**Scope:**

4: The work is relevant to the Web and to the track, and is of broad interest to the community

---

### Official Review · Reviewer_x9vp · 2023-11-24

**Novelty:** 6
**Technical Quality:** 5

**Review:**

Strength:
1.This paper proposes a new method for euphemism identification, which combines feature fusion and individualization to increase the semantic differences among euphemism and target keywords.
2.In the field of euphemism identification, the experimental results significantly exceed existing language models.

Weekness:
The model lacks a method for incremental updates.

**Questions:**

1.	As mentioned in the article, the number and meaning of euphemisms will continue to change with the changing network environment. Does your model have an incremental maintenance method?If a new drug and its euphemisms such as "sugar" appear in the future, how should you update your model?
2.	The input of the experiment is a sentence that already contains euphemisms of sensitive vocabulary. But I think the difficulty of this problem lies in how to distinguish whether a sentence contains euphemisms. For example, how can I determine if the word 'weed' in this sentence is related to agriculture or drug trading?I think this question should be more critical than which drug corresponds to 'weed'.

**Ethics Review Description:**

There are no ethical issues with this paper.

**Reviewer Confidence:**

3: The reviewer is confident but not certain that the evaluation is correct

**Scope:**

4: The work is relevant to the Web and to the track, and is of broad interest to the community

---

### Official Review · Reviewer_LTMP · 2023-11-24

**Novelty:** 5
**Technical Quality:** 6

**Review:**

This paper presents a new model for the task of identifying the meaning of euphemisms in natural language. The new model is based upon two key ideas: feature fusion, which aims to prevent similar contexts from confusing the meanings of euphemisms by "fusing" dynamic BERT-style contextual features with traditional static word embeddings, and feature individualization, which distinguishes semantically similar euphemisms by using orthogonal decomposition to eliminate the common dimensions between their feature representations. The paper finds that a classifier based upon the resulting modified features outperforms existing baselines, and also achieves performance that is better than or competitive with LLMs.

Pros:
- The proposed model is intuitively motivated, and the exact implementation of the feature fusion and individualization modules is described in a comprehensive and easy to understand way.
- Evaluation is comprehensive, including comparisons to existing baselines as well as more recent LLMs, and also including an ablation analysis that convincingly demonstrates the importance of the two novel modules.

Cons:
- Some presentation of results in unclear and could be improved. In particular, Table 5 purports to show a comparison between LLMs and FFI on the Drug, Weapon, and Sexuality datasets but does not actually say what metric the comparison is being done on (the reader has to turn to the main text to find that the metric is top1 accuracy, but this is easily missed)

Overall, I find this to be an interesting and novel approach to the problem of euphemism identification, and despite minor issues with presentation, I think this is a solid paper.

**Questions:**

I wanted to clarify what you see as the main novel contribution of this work: are feature fusion and feature individualization concepts that have been tried before on other tasks, with the main novelty here being that you combined them to do the euphemism task? Or are you claiming these as novel innovations in and of themselves?

**Reviewer Confidence:**

3: The reviewer is confident but not certain that the evaluation is correct

**Scope:**

3: The work is somewhat relevant to the Web and to the track, and is of narrow interest to a sub-community

---

### Decision · Program_Chairs · 2024-01-22

**Decision:**

Accept (Oral)

**Comment:**

In this work, the authors introduce a method for euphemism identification, enhancing the distinction between similar euphemisms in context and providing unique feature representations for target keywords, thereby outperforming state-of-the-art methods in accurately mapping euphemisms to their secret meanings on social networks and darknet marketplaces.

 The reviewers agree that this is an interesting problem, is novel, and the authors provide a useful first approach towards solving it. The evaluation on multiple datasets, including darknet marketplaces, is convincing. The reviewers (and authors) agree that the method is limited in its ability to identify novel euphemisms. I do think this paper represents a meaningful advance.